# Firing rate predictions in optimal balanced networks

**David G.T. Barrett**
Group for Neural Theory
École Normale Supérieure
Paris, France
david.barrett@ens.fr

**Sophie Denève**
Group for Neural Theory
École Normale Supérieure
Paris, France
sophie.deneve@ens.fr

**Christian K. Machens**
Champalimaud Neuroscience Programme
Champalimaud Centre for the Unknown
Lisbon, Portugal
christian.machens@neuro.fchampalimaud.org

## Abstract

How are firing rates in a spiking network related to neural input, connectivity and network function? This is an important problem because firing rates are a key measure of network activity, in both the study of neural computation and neural network dynamics. However, it is a difficult problem, because the spiking mechanism of individual neurons is highly non-linear, and these individual neurons interact strongly through connectivity. We develop a new technique for calculating firing rates in optimal balanced networks. These are particularly interesting networks because they provide an optimal spike-based signal representation while producing cortex-like spiking activity through a dynamic balance of excitation and inhibition. We can calculate firing rates by treating balanced network dynamics as an algorithm for optimising signal representation. We identify this algorithm and then calculate firing rates by finding the solution to the algorithm. Our firing rate calculation relates network firing rates directly to network input, connectivity and function. This allows us to explain the function and underlying mechanism of tuning curves in a variety of systems.

## 1 Introduction

The firing rate of a neuron is arguably the most important characterisation of both neural network dynamics and neural computation, and has been ever since the seminal recordings of Adrian and Zotterman [1] in which the firing rate of a neuron was observed to increase with muscle tension. A large, sometimes bewildering, diversity of firing rate responses to stimuli have since been observed [2], ranging from sigmoidal-shaped tuning curves [3, 4], to bump-shaped tuning curves [5], with much diversity in between [6]. What is the computational role of these firing rate responses and how are firing rates determined by neuron dynamics, network connectivity and neural input?

There have been many attempts to answer these questions, using a variety of experimental and theoretical techniques. However, most approaches have struggled to deal with the non-linearity of neural spike-generation mechanisms and the strong interaction between neurons as mediated through network connectivity. Significant progress has been made using linear approximations. For example, experimentally recorded firing rates in a variety of systems have been described using the linear receptive field, which captures the linear relationship between stimulus and firing rate response [7]. However, in recent years, it has been found that this linear approximation often fails to capture important aspects of neural activity [8]. Similarly, in theoretical studies, linear approximations

have been used to simplify non-linear firing rate calculations in a variety of network models, using Taylor Series approximations [9], and more recently, using linear response theory [10, 11]. These calculations have led to important insights into how neural network connectivity and input determine firing rates. Again, however, these calculations only apply to a restricted subset of situations, where the linearising assumptions apply.

We develop a new technique for calculating firing rates, by directly identifying the non-linear structure of tightly balanced networks. Balanced network theory has come to be regarded as the standard model of cortical activity [12, 13], accounting for a large proportion of observed activity through a dynamic balance of excitation and inhibition [14]. Recently, it was found that tightly balanced networks are synonymous with efficient coding, in which a signal is represented optimally subject to metabolic costs [15]. This observation allows us, here, to interpret balanced network activity as an optimisation algorithm. We can then directly identify that the non-linear relationship between firing rates, input, connectivity and neural computation is provided by this algorithm. We use this technique to calculate firing rates in a variety of balanced network models, thereby exploring the computational role and underlying network mechanisms of monotonic firing rate tuning curves, bump-shaped tuning curves and tuning curve inhomogeneity.

## 2   Optimal balanced network models

We calculate firing rates in a balanced network consisting of $N$ recurrently connected leaky integrate-and-fire neurons (Fig. 1a). The network is driven by an input signal $\mathbf{I} = (I_1, \ldots, I_k, \ldots I_M)$, where $I_k$ is the $k^{th}$ input and $M$ is the dimension of the input. In response to this input, neurons produce spike trains, denoted by $\mathbf{s} = (s_1, \ldots, s_i, \ldots, s_N)$, where $s_i(t) = \sum_k \delta(t - t_k^i)$ is the spike train of neuron $i$ with spike times $\{t_k^i\}$. A spike is produced whenever the membrane potential $V_i$ exceeds the spiking threshold $T_i$ of neuron $i$. This simple spike rule captures the essence of a neural spike-generation mechanism. The membrane potential has the following dynamics:

$$\frac{dV_i}{dt} = -\lambda V_i + \sum_{k=1}^{N} \Omega_{ik} s_k + \sum_{j=1}^{M} F_{ij} I_j \,, \tag{1}$$

where $\lambda$ is the neuron leak, $\Omega_{ik}$ is connection strength from neuron $k$ to neuron $i$ and $F_{ij}$ is the connection strength from input $j$ to neuron $i$ [16]. When a neuron spikes, the membrane potential is reset to $R_i \equiv T_i + \Omega_{ii}$. This is written in equation 1 as a self-connection. Throughout this work, we focus on networks where connectivity $\mathbf{\Omega}$ is symmetric - this simplifies our analysis, although in certain cases we can generalise to non-symmetric matrices.

We are interested in networks where a balance of excitation and inhibition coincides with optimal signal representation. Not all choices of network connectivity and spiking thresholds will give both [12, 13], but if certain conditions are satisfied, this can be possible. Before we proceed to our firing rate calculation, we must derive these conditions.

We begin by calculating the sum total of excitatory and inhibitory input received by neurons in our network. This is given by solving equation 1 implicitly:

$$V_i = \sum_{k=1}^{N} \Omega_{ik} r_k + \sum_{j=1}^{M} F_{ij} x_j \,, \tag{2}$$

where $r_k$ is a temporal filtering of the $k^{th}$ neuron's spike train

$$r_k = \int_0^\infty e^{-\lambda t'} s_k(t - t') \, dt' \,, \tag{3}$$

and $x_j$ is a temporal filtering of the $j^{th}$ input

$$x_j = \int_0^\infty e^{-\lambda t'} I_j(t - t') \, dt' \,. \tag{4}$$

All the excitatory and inhibitory inputs received by neuron $i$ are included in this summation (Eqn. 2). This can be rewritten as the slope of a loss function as follows:

$$V_i = -\frac{1}{2} \frac{dE(\mathbf{r})}{dr_i} \,, \tag{5}$$

where

$$E(\mathbf{r}) = -\mathbf{r}^T \mathbf{\Omega} \mathbf{r} - 2\mathbf{r}^T \mathbf{F} \mathbf{x} + c \qquad (6)$$

and c is a constant.

Now, we can use this expression to derive the conditions that connectivity must satisfy so that the network operates in an optimal balanced state. In balanced networks, excitation and inhibition cancel to produce an input that is the same order of magnitude as the spiking threshold. This is very small, relative to the magnitude of excitation or inhibition alone [12, 13]. In tightly balanced networks, which we consider, this cancellation is so precise that $V_i \to 0$ in the large network limit (for all active neurons) [15, 17, 18]. Now, using equation 5, we can see that this tight balance condition is equivalent to saying that our loss function (Eqn. 6) is minimised.

This has two implications for our choice of network connectivity and spiking thresholds. First, the loss function must have a minimum. To guarantee this, we require $-\mathbf{\Omega}$ to be positive definite. Secondly, the spiking threshold of each neuron must be chosen so that each spike acts to minimise the cost function. This spiking condition can be written as $E(\text{no spike}) > E(\text{with spike})$. Using equation 6, this can be rewritten as $E(\text{no spike}) > E(\text{no spike}) - 2[\mathbf{\Omega r}]_k - 2[\mathbf{Fx}]_k - \Omega_{kk}$. Finally,

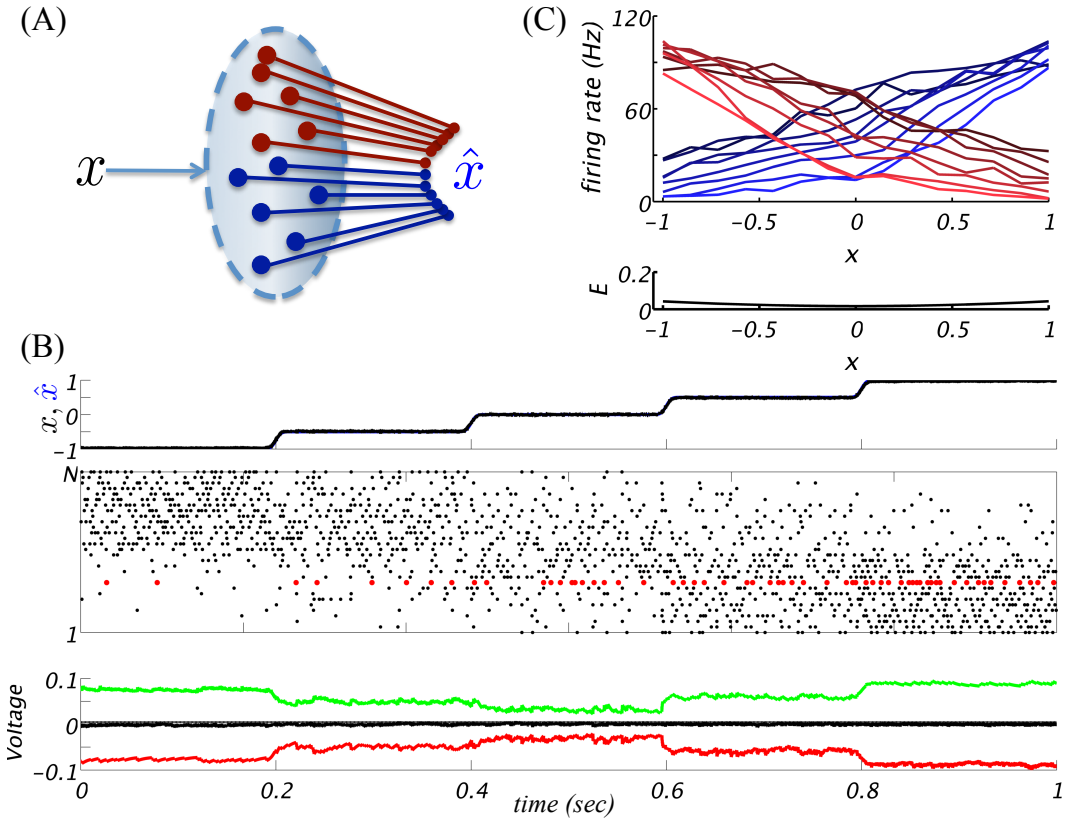

Figure 1: Optimal balanced network example. (A) Schematic of a balanced neural network providing an optimal spike-based representation $\hat{x}$ of a signal $x$. (B) A tightly balanced network can produce an output $\hat{x}_1$ (blue, top panel) that closely matches the signal $x_1$ (black, top panel). Population spiking activity is represented here using a raster plot (middle panel), where each spike is represented with a dot. For a randomly chosen neuron (red, middle panel), we plot the total excitatory input (green, bottom panel) and the total inhibitory input (red, bottom panel). The sum of excitation and inhibition (black, bottom panel) fluctuates about the spiking threshold (thin black line, bottom panel) indicating that this network is tightly balanced. A spike is produced whenever this sum exceeds the spiking threshold. (C) Firing rate tuning curves are measured during simulations of our balanced network. Each line represents the tuning curve of a single neuron. The representation error at each value of $x_1$ is given by equation 7.

cancelling terms, and using equation 2, we can write our spiking condition as $V_k > -\Omega_{kk}/2$. Therefore, the spiking threshold for each neuron must be set to $T_k \equiv -\Omega_{kk}/2$, though this condition can be relaxed considerably if our loss function has an additional linear cost term[1]. Once these conditions are satisfied, our network is tightly balanced.

We are interested in networks that are both tightly balanced and optimal. Now, we can see from equation 5 that the balance of excitation and inhibition coincides with the optimisation of our loss function (Eqn. 6). This is an important result, because it relates balanced network dynamics to a neural computation. Specifically, it allows us to interpret the spiking activity of our tightly balanced network as an algorithm that optimises a loss function (Eqn. 6).

This is interesting because this optimisation can be easily mapped onto many useful computations. A particularly interesting example is given by $\mathbf{\Omega} = -\mathbf{F}\mathbf{F}^T - \beta\mathbf{I}$, where $\mathbf{I}$ is the identity matrix [15, 17, 18]. In recent work, it was shown that this connectivity can be learnt using a spike timing-dependent plasticity rule [15]. Here, we use this connectivity to rewrite our loss function (Eqn. 6) as follows:

$$E = (\mathbf{x} - \hat{\mathbf{x}})^2 + \beta\sum_{i=1}^{N} r_i^2 \,, \tag{7}$$

where

$$\hat{\mathbf{x}} = \mathbf{F}^T\mathbf{r} \,. \tag{8}$$

The second term of equation 7 is a metabolic cost term that penalises neurons for spiking excessively, and the first term quantifies the difference between the signal value $\mathbf{x}$ and a linear read-out, $\hat{\mathbf{x}}$, where $\hat{\mathbf{x}}$ is computed using the linear decoder $\mathbf{F}^T$ (Eqn. 8). Therefore, a network with this connectivity produces spike trains that optimise equation 7, thereby producing an output $\hat{\mathbf{x}}$ that is close to the signal value $\mathbf{x}$. Throughout the remainder of this work, we will focus on optimal balanced networks with this form of connectivity.

We illustrate the properties of this system by simulating a network of 30 neurons. We find that our network produces spike trains (Fig. 1 b, middle panel) that represent $\mathbf{x}$ with great accuracy, across a broad range of signal values (Fig. 1 b, top panel). As expected, this optimal performance coincides with a tight balance of excitation and inhibition (Fig. 1 b, bottom panel), reminiscent of cortical observations [14]. In this example, our network has been optimised to represent a 2-dimensional signal $\mathbf{x} = (x_1, x_2)$. We measure firing rate tuning curves using a fixed value of $x_2$ while varying $x_1$. We use this signal because it can produce interesting, non-linear tuning curves (Fig. 1 c), especially at signal values where neurons fall silent. In the next section, we will attempt to understand this tuning curve non-linearity by calculating firing rates analytically.

## 3 Firing rate analysis with quadratic programming

Our goal is to calculate the firing rates $\mathbf{f}$ of all the neurons in these tightly balanced network models as a function of the network input, the recurrent network connectivity $\mathbf{\Omega}$, and the feedforward connectivity $\mathbf{F}$. On the surface, this may seem to be a difficult problem, because individual neurons have complicated non-linear integrate-and-fire dynamics and they interact strongly through network connectivity. However, the loss function relationship that we developed above allows us now to circumvent these problems.

There are many possible firing rate measures used in experiments and theoretical studies. Usually, a box-shaped temporal averaging window is used. We define the firing rate of a neuron to be:

$$f_k = \lambda \int_0^\infty e^{-\lambda t'} s_k(t - t') \, dt' \,. \tag{9}$$

This is an exponentially weighted temporal average[2], with timescale $\lambda^{-1}$. We have chosen this temporal average because it matches the dynamics of synaptic filters in our neural network (Eqn. 3),

allowing us to write $f_i(t) = \lambda r_i(t)$. Here, we need to multiply by $\lambda$ to ensure that our firing rates are reported in units of spikes per second.

We can now calculate firing rates using this relationship and by exploiting the algorithmic nature of tightly balanced networks. These networks produce spike trains that minimise our loss function $E(\mathbf{r})$ (Eqn. 6). Therefore, the firing rates of our network are those that minimise $E(\mathbf{f}/\lambda)$, under the constraint that firing rates must be positive:

$$\{f_i\} = \arg\min_{f_i \geq 0} E(\mathbf{f}/\lambda). \tag{10}$$

This firing rate prediction is the solution to a constrained optimisation problem known as quadratic programming [19]. The optimisation is quadratic, because our loss function is a quadratic function of $\mathbf{f}$, and it is constrained because firing rates are positive valued quantities, by definition.

We illustrate this firing rate prediction using a simple two-neuron network, with recurrent connectivity given by $\mathbf{\Omega} = -\mathbf{F}^T\mathbf{F} - \beta\mathbf{I}$ as before. We simulate this system and measure the spike-train firing rates for both neurons (Fig. 2 a, left panel). We then use equation 10 to obtain a theoretical prediction for firing rates. We find that our firing rate prediction matches the spike-train measurement with great accuracy (Fig.2 a, middle panel and right panel).

We can now use our firing rate solution to understand the relationship between firing rates, input, connectivity and function. When both neurons are active, we can solve equation 10 exactly, to see that firing rates are related to network connectivity according to $\mathbf{f} = -\lambda\mathbf{\Omega}^{-1}\mathbf{F}\mathbf{x}$. When one of the neurons becomes silent, the other neuron must compensate by adjusting its firing rate slope. For example, when neuron 1 becomes silent, we have $f_1 = 0$ and the firing rate of neuron 2 increases to $f_2 = \lambda\mathbf{F}_2\mathbf{x}/(\mathbf{F}_2\mathbf{F}_2^T + \beta\mathbf{I})$, where $\mathbf{F}_2$ denotes the second row of $\mathbf{F}$. Similarly, when neuron 2

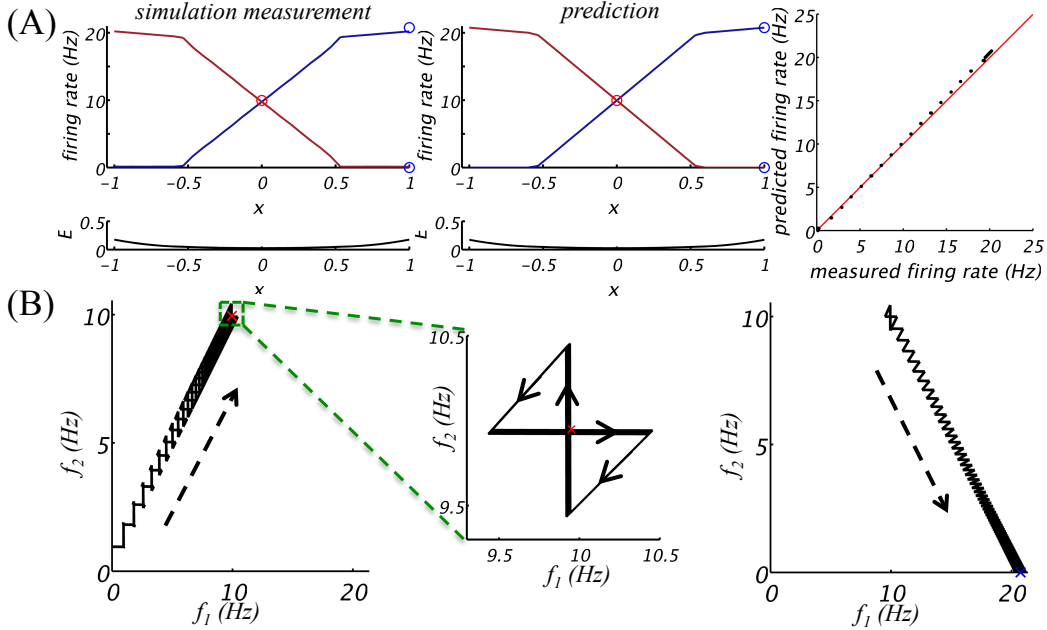

Figure 2: Calculating firing rates in a two-neuron example. (A) Tuning curve measurements are obtained from a simulation of a two-neuron network (left, top). The representation error $E$ for this network is given at each signal value $x$ (left, bottom). Tuning curve predictions are obtained using quadratic programming (middle, top), with predicted representation error $E$ (middle, bottom). Predicted firing rates closely match measured firing rates for both neurons, and for all signal values (right). (B) A phase diagram of the network activity during a simulation (left panel). Firing rates evolve from a silent state towards the minimum of the cost function $E(x_1 = 0)$ (red cross, left panel). Here, they fluctuate about the minimum, increasing in discrete steps of size $\lambda$ and decreasing exponentially (left panel, inset).We also measure the firing rate trajectory (right panel) as the network evolves towards the minimum of the cost function $E(x_1 = 1)$ (blue cross, right panel), where neuron 2 is silent.

becomes silent, we have $f_2 = 0$, and the firing rate of neuron 1 increases to $f_1 = \lambda \mathbf{F}_1 \mathbf{x}/(\mathbf{F}_1 \mathbf{F}_1^T + \beta \mathbf{I})$, where $\mathbf{F}_1$ is the first row of $\mathbf{F}$. This non-linear change in firing rates is caused by the positivity constraint. It can be understood functionally, as an attempt by the network to represent $\mathbf{x}$ accurately, within the constraints of the system.

In larger networks, our firing rate prediction is more difficult to write down analytically because there are so many interactions between individual neurons and the positivity constraint. Nonetheless, we can make a number of general observations about tuning curve shape. In general, we can interpret tuning curve shape to be the solution of a quadratic programming problem, which can be written as a piece-wise linear function $\mathbf{f} = \mathbf{M}(\mathbf{x}) \cdot \mathbf{x}$, where $\mathbf{M}(\mathbf{x})$ is a matrix whose entries depend on the region of signal space occupied by $\mathbf{x}$. For example, in the two-neuron system that we just discussed, the signal space is partitioned into three regions: one region where neuron 1 is active and where neuron 2 is silent, a second region where both neurons are active and a third region where neuron 1 is silent and neuron 2 is active (Fig. 2 a, left panel). In each region there is a different linear relationship between the signal and the firing rates. The boundaries of these regions occur at points in signal space where an active neuron becomes silent (or where a silent neuron becomes active). At most, there will be $N + 1$ such regions.

We can also use quadratic programming to describe the spiking dynamics underlying these non-linear networks. Returning to our two-neuron example, we measure the temporal evolution of the firing rates $f_1$ and $f_2$. We find that if we initialise the network to a sub-optimal state, the firing rates rapidly evolve toward the optimum in a series of discrete steps of size $\lambda$ (Fig. 2 b, left panel). The step-size is $\lambda$ because when neuron $i$ spikes, $r_i \rightarrow r_i + 1$, according to equation 3, and therefore, $f_i \rightarrow f_i + \lambda$, according to equation 9. Once the network has reached the optimal state, it is impossible for it to remain there. The firing rates begin to decay exponentially, because our firing rate definition is an exponentially weighted summation (Eqn. 9) (Fig. 2 b, middle panel). Eventually, when the firing rate has decayed too far from the optimal solution, another spike is fired and the network moves closer to the optimum. In this way, spiking dynamics can be interpreted as a quadratic programming algorithm. The firing rate continues to fluctuate around the optimal spiking value. These fluctuations are noisy, in that they are dependent on initial conditions of the network. However, this noise has an unusual algorithmic structure that it is not well characterised by standard probabilistic descriptions of spiking irregularity.

## 4 Analysing tuning curve shape with quadratic programming

Now that we have a framework for relating firing rates to network connectivity and input, we can explore the computational function of tuning curve shapes and the network mechanisms that generate these tuning curves. We will investigate systems that have monotonic tuning curves and systems that have bump-shaped tuning curves, which together constitute a large proportion of firing rate observations [2, 3, 4, 5].

We begin by considering a system of monotonic tuning curves, similar to the examples that we have considered already where recurrent connectivity is given by $\mathbf{\Omega} = -\mathbf{F}\mathbf{F}^T - \beta \mathbf{I}$. In these systems, the recurrent connectivity and hence the tuning curve shape is largely determined by the form of the feedforward matrix $\mathbf{F}$. This matrix also determines the contribution of tuning curves to computational function, through its role as a linear decoder for signal representation (Eqn. 8). We illustrate this by simulating the response of our network to a 2-dimensional signal $\mathbf{x} = (x_1, x_2)$, where $x_1$ is varied and $x_2$ is fixed, using three different configurations of $\mathbf{F}$ (Fig. 3). This system produces monotonically increasing and decreasing tuning curves (Fig. 3a). We find that neurons with positive values of $\mathbf{F}$ have positive firing rate slopes (Fig. 3, blue tuning curves), and neurons with negative $\mathbf{F}$ values have negative firing rate slopes (Fig. 3, red tuning curves). If the values of $\mathbf{F}$ are regularly spaced, then the tuning curves of individual neurons are regularly spaced, and, if we manipulate this regularity by adding some random noise to the connectivity, we obtain inhomogeneous and highly irregular tuning curves (Fig.3 b). This inhomogeneity has little effect on the representation error.

This inhomogeneous monotonic tuning is reminiscent of tuning in many neural systems, including the oculomotor system [4]. The oculomotor system represents eye position, using neurons with negative slopes to represent left side eye positions and neurons with positive slopes to represent right side eye positions. To relate our model to this system, the signal variable $x_1$ can be interpreted as eye-position, with zero representing the central eye position, and with positive and negative values

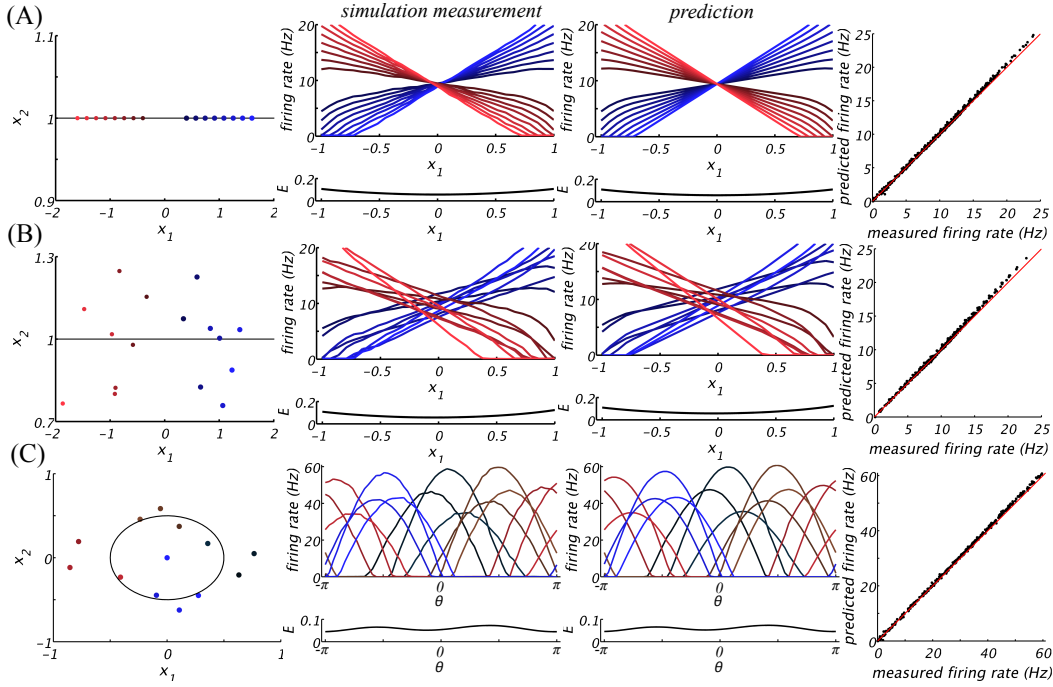

Figure 3: The relationship between firing rates, stimulus and connectivity in a network of 16 neurons. (A) Each dot represents the contribution of a neuron to a signal representation (when the firing rate is $10 \times 16$ Hz) ($1^{st}$ column). Here, we consider signals along a straight line (thin black line). We simulate a network of neurons and measure firing rates ($2^{nd}$ column). These measurements closely match our algorithmically predicted firing rates ($3^{rd}$ column), where each point in the 4th column represents the firing rate of an individual neuron for a given stimulus. (B) Similar to '(A)' except that some noise is added to the connectivity. The representation error (bottom panels, column 2 and column 3) is similar to the network without connectivity noise. (C) Similar to '(B)', except that we consider signals along a circle (thin black line). Each dot represents the contribution of a neuron to a signal representation (when the firing rate is $20 \times 16$ Hz) ($1^{st}$ column). This signal produces bump-shaped tuning curves ($2^{nd}$ column), which we can also predict accurately ($3^{rd}$ and $4^{th}$ column).

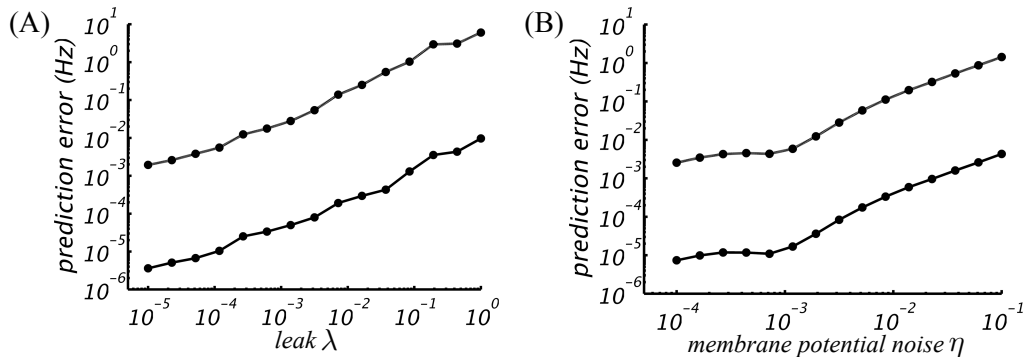

Figure 4: Performance of quadratic programming in firing rate prediction. (A) The mean prediction error (absolute difference between each prediction and measurement, averaged over neurons and over 0.5 seconds) increases with $\lambda$ (bottom line). The standard deviation of the prediction becomes much larger with $\lambda$ (top line). (B) The mean prediction error (bottom line) and standard deviation of the prediction error (top line) also increase with noise. However, the prediction error remains less that 1 Hz.

of $x_1$ representing right and left side eye positions, respectively. Now, we can use the relationship that we have developed between tuning curves and computational function to interpret oculomotor tuning as an attempt to represent eye positions optimally.

Bump-shaped tuning curves can be produced by networks representing circular variables $x_1 = \cos\theta$, $x_2 = \sin\theta$, where $\theta$ is the orientation of the signal (Fig. 3 c). As before, the tuning curves of individual neurons are regularly spaced if the values of $\mathbf{F}$ are regularly spaced. If we add some noise to the connectivity $\mathbf{F}$, the tuning curves become inhomogeneous and highly irregular. Again, this inhomogeneity has little effect on the signal representation error.

In all the above examples, our firing rate predictions closely match firing rate measurements from network simulations (Fig. 3). The success of our algorithmic approach in calculating firing rates depends on the success of spiking networks in algorithmically optimising a cost function. The resolution of this spiking algorithm is determined by the leak $\lambda$ and membrane potential noise. If $\lambda$ is large, the firing rate prediction error will have large fluctuations about the optimal firing rate value (Fig. 4 a). However, the average prediction error (averaged over time and neurons) remains small. Similarly, membrane potential noise[3] increases fluctuations about the optimal firing rate but the average prediction error remains small (until the noise is large enough to generate spikes without any input) (Fig. 4 b).

## 5 Discussion and Conclusions

We have developed a new algorithmic technique for calculating firing rates in tightly balanced networks. Our approach does not require us to make any linearising approximations. Rather, we directly identify the non-linear relationship between firing rates, connectivity, input and optimal signal representation. Identifying such relationships is a long-standing problem in systems neuroscience, largely because the mathematical language that we use to describe information representation is very different to the language that we use to describe neural network spiking statistics. For tightly balanced networks, we have essentially solved this problem, by matching the firing rate statistics of neural activity to the structure of neural signal representation. The non-linear relationship that we identify is the solution to a quadratic programming problem.

Previous studies have also interpreted firing rates to be the result of a constrained optimisation problem [21], but for a population coding model, not for a network of spiking neurons. In a more recent study, a spiking network was used to solve an optimisation problem, although this network required positive and negative spikes, which is difficult to reconcile with biological spiking [22].

The firing rate tuning curves that we calculate have allowed us to investigate poorly understood features of experimentally recorded tuning curves. In particular, we have been able to evaluate the impact of tuning curve inhomogeneity on neural computation. This inhomogeneity often goes unreported in experimental studies because it is difficulty to interpret [6], and in theoretical studies, it is often treated as a form of noise that must be averaged out. We find that tuning curve inhomogeneity is not necessarily noise because it does not necessarily harm signal representation. Therefore, we propose that tuning curves are inhomogeneous simply because they can be.

Beyond the interpretation of tuning curve shape, our quadratic programming approach to firing rate calculations promises to be useful in other areas of neuroscience - from data analysis, where it may be possible to train our framework using neural data so as to predict firing rate responses to sensory stimuli - to the study of computational neurodegeneration, where the impact of neural damage on tuning curves and computation may be characterised.

**Acknowledgements**

We would like to thank Nuno Calaim for helpful comments on the manuscript. Also, we are grateful for generous funding from the Emmy-Noether grant of the Deutsche Forschungs-gemeinschaft (CKM) and the Chaire dexcellence of the Agence National de la Recherche (CKM, DB), as well as a James Mcdonnell Foundation Award (SD) and EU grants BACS FP6-IST-027140, BIND MECT-CT-20095-024831, and ERC FP7-PREDSPIKE (SD).

## Footnotes

[1] Suppose that our network optimises the following cost function: $E(\mathbf{r}) = -\mathbf{r}^T\mathbf{\Omega}\mathbf{r} - 2\mathbf{r}^T\mathbf{F}\mathbf{x} + c + \mathbf{b}^T\mathbf{r}$, where $\mathbf{b}$ is a vector of positive linear weights. Then, we find that the optimal spiking thresholds for this network are given by $T_i \equiv (-\Omega_{ii} + b_i)/2 \geq -\Omega_{ii}/2$. Therefore, we can apply our techniques to all networks with thresholds $T_i \geq -\Omega_{ii}/2$.

[2] In this case, the firing rate timescale is very short, because $\lambda$ is the membrane potential leak. However, we can easily generalise our framework so that this timescale can be as long as the slowest synaptic process [17, 18].

[3]Membrane potential noise can be included in our network model by adding a Wiener process noise term to our membrane potential equation (Eqn. 1). We parametrise this noise with a constant $\eta$.

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
