[Reviews · NeurIPS 2013]

Submitted by Assigned_Reviewer_5

A basic question in neuroscience concerns the relation between network structure and function. A major difficulty in relating the two is multi-leveled complexity of neural systems in terms on nonlinearity, variability and stochasticity. The present is concerned with networks of spiking neurons which attempt to optimally represent time-varying input signals. The performance cost function comprises a quadratic error function and a penalty for high activity. The authors show how the natural network dynamics of a network of spiking neurons can be described as minimizing this cost function, leading to a so-called balanced state, which is, arguably, widely regarded as a natural state for biological neuron dynamics. An important attribute of the present formulation is that it does not rely on linear approximations, as has often been used in the past, leading to a more faithful representation of the true dynamics. Within this framework the authors compute the firing rates as the solution of a quadratic problem subject to non-negativity constraints on the rates, allowing them to express the solution in closed form as piece-wise linear function. They also relate the types of tuning curves obtained (monotonic or bell-shaped) to the nature of the input, and show that optimal performance can be achieved using heterogeneous tuning curves, as has been, indeed, observed in biological systems.
Overall I found this paper interesting and insightful, and, in combination with ref. [13] that deals with synaptic plasticity, constitute an original contribution to the structure-function relationship.
Specific comments:
1) The form of E in (6) suggests that W must be symmetric (since the anti-symmetric component of W does not contribute to the quadratic term). However, eq. (7) itself does not seem to require symmetry. Please relate to this issue.
2) Equation (6) suggests that it must have a minimum at which dE/d\nu=0 so that \dot V = 0. However, why does this entail V_i=0 (balanced state) rather than V_i=constant.
3) Typo: Line 330, where $\beta I$ - something is missing.
Summary: An interesting paper motivating the balanced state in spiking neural networks as a solution to an optimal representation problem. This is an original and insightful contribution.

Submitted by Assigned_Reviewer_6

This paper presents a method to compute the firing rates in recurrent networks of integrate and fire neurons for given inputs by imposing the condition of optimal balance of excitation and inhibition. While rather technical, the paper is well written, the approach is new, interesting and seems to be sound. The results are convincing, except maybe that the spiking (as shown in Fig. 1) does not appear as stochastic as in cortex. Taken together the approach has a huge potential to foster our understanding of the operation state and the function of cortical networks involved in sensory processing.
Summary: This paper presents a new method to compute the firing rates in recurrent networks of integrate and fire neurons for given inputs by imposing the condition of optimal balance of excitation and inhibition

Submitted by Assigned_Reviewer_7

This is an interesting paper that describes how the firing rates in a balanced network represent solutions to a quadratic cost function reflecting deviations between input signals and decoded output. The paper is well written aside from a few points (described below) that need clarification.

Unfortunately, however, this paper represents only a relatively small conceptual advance over Ref. 13 (Bourdoukan et al., NIPS 2012) as well as the paper under revision submitted as a supplement. Specifically, the entire section 2 here was already described by Bourdoukan et al NIPS 2012. The fact that an inhomogeneous network can represent signals with a performance similar to those exhibited by homogeneous systems was made by Bourdoukan et al as well as in a number of other recent neuroscience publications. This leaves only the calculation of the firing rates as a new contribution. Nevertheless, this too was essentially done by Bourdoukan et al 2012 NIPS (including the condition for spiking V_i > T_i).

One aspect that this paper also needs to be address is that the proposed balanced networks have a very specific structure – the number of inputs equals the number of neurons in the recurrent network. Therefore, it is not clear how one may consider divergence and convergence that often are hallmark features of neural circuits.

Minor comments that need to be clarified:

-- Eq. 3, I think the lower integration limit has to be 0 ( the upper integration limit may extend to infinity)

-- Figure 1 legend, “A spike is produced whenever the total input exceeds the spiking threshold (thin black line, bottom panel). However the number of spikes does not match the number of red dots in the upper panels that are also described as marking spike times from this neuron.

-- Line 187 “We find that these curves are highly nonlinear” –The curves in Figure 1c look quite linear.
Summary: This paper is nicely written and is quite interesting when read as a stand alone contribution. However, the conceptual advance relatively to Ref. 13 is questionable.
Author Feedback

Author rebuttal: We would like to begin by thanking the referees for reviewing our work.

Clearly, reviewer 7 is in disagreement with reviewers 5 and 6, who have both argued strongly for acceptance of our paper. We disagree with the arguments proposed by reviewer 7. In particular, reviewer 7 claims that the primary contribution of our work (a new method to calculate firing rates using quadratic programming), was previously published by Bourdoukan et. al. 2012 (Ref. 13). This is not the case. There is no mention of quadratic programming in that work, and all firing rates reported in that work are measured in neural network simulations. In fact, our work partly grew out of a desire to understand the firing rate tuning curves measured in the simulations of Bourdoukan et. al and other similar balanced network models. Indeed, we cite this as an important motivation in our introduction. Furthermore, this view is supported by reviewer 5, who "found this paper interesting and insightful, and, in combination with ref. [13] that deals with synaptic plasticity, constitute an original contribution to the structure-function relationship."

We will now address each reviewers questions, before returning to this issue:

Reviewer 5:

1) Equation 7 does require symmetry because in this case W_ij=\Gamma_i \Gamma_j + \beta \delta_{ij} and this is a symmetric matrix.
2) If dE/d\nu=0 then V_i=0, which does correspond to the balanced state (see line 147)
3) We believe that this work has the potential to have a major impact on the neuroscience subset of the NIPS community. Indeed, since our NIPS submission, a number of experimental neuroscience research groups have begun to apply our firing rate calculation method to data analysis. This suggests that the impact of this work will extend far beyond the neural network dynamics community.

Reviewer 6:

1) The regularity of spike trains in Figure 1 is the result of our choice of parameters. We have found that there are many alternative set of parameters that can produce greater levels of cortex-like irregularity (such as small \beta). We will use these for our final submission.

Reviewer 7:

1) "The entire section 2 here was already described by Bourdoukan et al NIPS 2012."

We disagree with this. The first four paragraphs of section 2 describe a generic integrate and fire model. The next two paragraphs relate membrane potentials directly to a cost function (a different cost function to that used in Bourdoukan et al.). Only in the last four paragraphs do we develop the relationship between our work and Bourdoukan et al. We do this deliberately because this is a particularly interesting example of our general framework, and has attracted huge attention during the last year. We could have omitted the last 3 paragraphs in this section and simply referred to Bourdoukan et al. Instead, however, we described their work (with extensive referencing) so that our paper may be read as a coherent whole, without the need to read other papers first.

2) "This leaves only the calculation of the firing rates as a new contribution. Nevertheless, this too was essentially done by Bourdoukan et al 2012 NIPS"

This is not the case: Bourdoukan et al do not calculate firing rates. There is no mention of quadratic programming, or the properties of this algorithm in Bourdoukan et al. Indeed, the ambiguity of the relationship between firing rates and stimulus in Bourdoukan et al (amongst others) was one of the key motivations in our study, as we mention in the introduction.

Also, we believe that we have contributed more than "only the calculation of firing rates". In particular, we have explained exactly how balanced spiking networks can generate bump-shaped tuning curves and oculomotor-like tuning curves. We have also explained the origin of tuning curve diversity.

Perhaps the confusion here is that Bourdoukan et al. (and the authors of the attached preprint) discuss firing rates in the same network model that we study. The difference between our work and the work of Bourdoukan et al. is that we analytically calculate firing rates (using quadratic programming) whereas they measure firing rates in a simulation of a spiking network. The problem with simulation measurements is that they often cannot reveal the exact non-linear relationship between a neural input and network firing rates. Our key insight was to identify this relationship directly. We found that is is determined by a quadratic programming. In this way we could understand tuning curve shape in the model introduced by Bourdoukan et al. and in other balanced network models.

3) "One aspect that this paper also needs to be address is that the proposed balanced networks have a very specific structure – the number of inputs equals the number of neurons in the recurrent network. Therefore, it is not clear how one may consider divergence and convergence that often are hallmark features of neural circuits."

The number of inputs is not equal to the number of neurons. This is clearly stated in lines 69 and 70 of the text, where we explain that there may be N neurons in our network and M inputs. Indeed, in all our examples, the dimension of the input is M=2. As such there is an extremely large divergence from the input to the network.


Again, we would like to thank the referees for reviewing our work.